# Spatio-Temporal Difference in Agricultural Eco-Efficiency and Its Influencing Factors Based on the SBM-Tobit Models in the Yangtze River Delta, China

**DOI:** 10.3390/ijerph20064786

**Published:** 2023-03-08

**Authors:** Lin Shi, Xiaofei Shi, Fan Yang, Lixue Zhang

**Affiliations:** School of Economics and Management, Zhejiang Ocean University, Zhoushan 316022, China; shilin@zjou.edu.cn (L.S.); shixiaofei@zjou.edu.cn (X.S.); yang-fan@zjou.edu.cn (F.Y.)

**Keywords:** Yangtze River Delta region, agricultural eco-efficiency, SBM-Tobit model, carbon emissions

## Abstract

In the Yangtze River Delta region, where the agricultural economy is well developed and agricultural pollution and carbon emissions are significant, a regional study of AEE (Agricultural Eco-Efficiency) is crucial to reducing agricultural environmental pollution, improving the rationalization of agricultural production layout, and promoting the realization of low-carbon goals. The SBM-Tobit model and GIS were employed to analyze AEE based on the carbon emission evaluation system in the spatial and temporal characteristics, as well as the influencing factors and the migration path of the center of gravity in the “low carbon” context. A rational plan of agricultural production was proposed according to the results. The following results were obtained: (1) The level of AEE in the Yangtze River Delta region was high, and the region exhibited a U-shaped curve change from 2000 to 2020, with a fluctuating decrease from 2000 to 2003 and a fluctuating increase from 2004 to 2020. The regional spatial development balance was enhanced, while there was a spatial incongruity in the development process of AEE enhancement, high in the southwest and low in the northeast; (2) AEE generally had a high regionalized agglomeration of low–low in the southwest and high–high in the northeast. Nonetheless, temporal heterogeneity was observed in spatial correlation, and the correlation weakened with time variation; (3) Urbanization level, agricultural production structure, crop cultivation structure, and fertilizer application intensity were the main factors influencing AEE in the Yangtze River Delta region; (4) The center of gravity of AEE in the Yangtze River Delta region shifted to the southwest under the influence of “low-carbon” related policies. Therefore, the improvement of AEE in the Yangtze River Delta region should focus on inter-regional coordination and linkages, rational planning of production factors, and the formulation of measures under relevant carbon policies.

## 1. Introduction

Eco-efficiency is an effective tool to measure the degree of coordination between socio-economic development and environmental impact [1,2], which can be measured by the ratio of increased socio-economic value to environmental impact [3]. AEE (Agricultural Eco-Efficiency) indicates the reduction in agricultural factor inputs and resource consumption, as well as the reduction in environmental pollution under the premise of ensuring output [4,5]. It is the application of the eco-efficiency concept in the field of agriculture. Although China is developing rapidly in the agricultural field, the traditional crude agricultural economy has brought about severe environmental problems such as high consumption and high emissions [6,7]. The surface source pollution caused by agricultural production has put more pressure on China’s agricultural environment, and resource consumption and pollutant emissions have almost reached the environmental carrying limit [6,8]. The green and sustainable development of agricultural production is an essential national solution to these problems. In turn, AEE is one of the basic factors for the improvement of ecosystem services and the sustainability of agricultural production [4]. A reasonable assessment and analysis of regional AEE is beneficial for the mastery of agricultural production and provides relevant theoretical data for the sustainable development planning of agriculture.

With the introduction of green agriculture, a growing number of researchers began to investigate the issue of AEE, among which the early ones focused on AEE evaluation methods, spatial and temporal differences, and the exploration of constraints [3,5]. Since 2000, the development trend of AEE in China has been stable but with significant regional differences, and technical heterogeneity is a key factor contributing to its variability [9]. With the gradual depth of research, multi-scale agroecological efficiency measurement and influence factors exploration have been widely researched [10,11]. Even though the agroecological efficiency in China is on a growing trend, the development is extremely uneven among regions, with the highest efficiency in the eastern region, followed by the central and western regions, and the lowest in the northeastern region [12]. This is primarily impacted by the factors of agricultural industry structure and agricultural technology level [13,14]. Additionally, some researchers have emphasized the relationship between AEE and agricultural economic development [10], endogenous growth mechanisms and exogenous ways to improve [15], spatial scale effect and convergence of AEE [16], and AEE multi-perspective development [17]. Concurrently, the construction of an agricultural eco-efficiency value system demonstrates diversity, while most researchers take agricultural production related factors [18,19] as desired outputs, as well as carbon emission [3,20] and surface source pollution [8,21] as non-desired outputs. Some aspects have not been covered. For example, first, the construction of the current agricultural eco-efficiency system is still deficient, the composition of desired outputs in the study is complex, and agricultural ecosystem services involve human well-being [22], which can be used as part of the desired outputs in the system. Second, the research scale generally takes the provincial level as the decision-making unit [18], and there are few reports on the study of AEE at the prefectural level [15]. Thirdly, the spatial and temporal evolution of AEE has been rarely studied [23], especially in the context of the “double carbon” policy, and the trend of the shift of focus remains unknown.

Currently, the methods for measuring agroecological efficiency include the Single Indicator and Multi-indicator Comprehensive System Method [24,25], Hyperbolic Distance Function Method [1], Ratio Method [26], Life Cycle Assessment [5], Stochastic Frontier Analysis [27,28], and Data Envelopment Analysis with multiple inputs and outputs [3,29]. Among them, Super-SBM in data envelopment analysis has become the main method to study AEE. Its advantages overcome the shortcomings of single-indicator analysis (which cannot distinguish between different environmental impacts) and multi-indicator analysis (which has subjective factors in determining weights) [18,30]. Meanwhile, adding non-desired outputs solves the slack problem of traditional DEA that neglect inputs and outputs [31,32]. Additionally, researchers have selected non-desired outputs for agroecological efficiency in the model mainly for agricultural water pollution and agricultural chemical pollution [6,9]. However, agricultural production generates a large amount of carbon every year due to the allocation and storage of machinery operations, tillage, irrigation, and fertilizers [3,10]. It is also a major source of significant negative environmental impacts [33]. Hence, the inclusion of agricultural carbon emissions as non-desired outputs in the indicator system is necessary for the accuracy of AEE values. In recent years, China has increased its policy on carbon emission reduction to achieve “peak carbon” by 2030 and “carbon neutral” by 2060 [34]. The improvement of AEE, as one of the vital factors affecting the achievement of carbon emission reduction goals [20], is a realistic requirement for the green and sustainable development of agriculture and a critical way to achieve carbon emission reduction [20,35].

The Yangtze River Delta region is one of the regions with the earliest rural reform, the most comprehensive agricultural industrial system, the strongest agricultural science and technology innovation capability, and the fastest urban–rural integration process in China. It occupies a pivotal position in the overall situation of national agricultural and rural modernization construction. The Yangtze River Delta region is economically active, gathers 16% of China’s population, and occupies nearly a quarter of the total economic volume, causing huge consumption of environmental resources and energy in the region, as well as outstanding structural pollution and high pressure on carbon emission attainment. Thus, the assessment of the regional AEE level can lay a theoretical foundation for the green and sustainable development of agriculture and the further realization of carbon emission reduction goals. Given the above issues, the AEE in the Yangtze River Delta region was assessed through the Super-SBM model with agricultural carbon emissions as a non-desired output indicator. Meanwhile, GIS was further adopted to spatialize regional AEE and enhance the visualization of AEE status in the Yangtze River Delta region. The analysis of the spatial aggregation of AEE and the shift of the center of gravity under the policy, as well as the assessment of the spatial correlation within each region and the effect of changes in efficiency values after low carbon development, can support the formulation and implementation of regional agricultural production policies. Furthermore, the impact factors of the Yangtze River Delta region were evaluated by the Tobit model, providing a theoretical basis for green and sustainable management of regionalized agriculture.

The paper focuses on the spatial and temporal characteristics, driving mechanisms, the shift of focus, and related policy aspects of eco-efficiency in the Yangtze River Delta, calculated with a gross value of agricultural production and agroecosystem services as desired outputs and carbon emissions as non-desired outputs. The contributions of this study are summarized as follows.

The innovative AEE indicator system is a new combination of agricultural factor inputs and agricultural outputs, introducing agroecological system service indicators into desired outputs to compensate for the neglected part of the gross value of agricultural production and reinforcing the accuracy of AEE values in the region.The areas of incompatible agricultural production and environmental protection in the region are identified, and construction advice is proposed. Specifically, the spatial and temporal characteristics, as well as driving mechanisms of agricultural eco-efficiency in the region, are assessed. The areas of slow efficiency development are identified, and relevant targeted suggestions for problem areas are provided.The effect of the “double carbon” policy is effectively verified. The migration of agricultural eco-efficiency pathways under the “double carbon” policy reflects the impact of related policies.

## 2. Literature Review

Agriculture is a crucial socio-economic sector for sustainable social development and climate change mitigation [1,3,36]. It is one of the main factors that cannot be ignored in assessing the environmental sustainability of different crops owing to the large number of greenhouse gases produced [5,37]. China is a large agricultural country, and the development of its agriculture provides an essential guarantee for China’s economic and social stability [11,15]. However, a series of ecological and environmental problems, such as the consumption of limited natural and energy resources and the increased release of greenhouse gases, emerge when agricultural growth is mainly generated by the intensive input of production factors [6,10]. Among them, crop production for human consumption generates more than 21% of greenhouse gas emissions, equivalent to 2.8 Gt of carbon dioxide, which severely constrains sustainable agricultural development [7,12]. Thus, researchers, governments, and organizations are studying the energy cost, production, and output of agriculture at a deeper level to improve the sustainable management of agriculture. Roberto measured the carbon emissions of crop production processes through CF and curtailed greenhouse gas emissions in a circular economy model [7]. Cathal optimized agricultural cost energy by modeling agricultural production energy using the renewable energy value chain from an economic and environmental perspective [36]. Mónica promoted sustainable public management systems by replacing crop production systems with sustainable and circular production models [37]. Additionally, the EU Green Deal established a “farm-to-table” sustainable agricultural development strategy to reinforce equitable, healthy, and environmentally friendly agricultural systems to wrestle with climate change [35,38]. Sustainable production in agricultural systems is critical to handling societal concerns about environmental impacts and nutritional values while maintaining economically viable production systems for farmers. Its combined importance is manifested in satisfying human needs for food and enhancing environmental, economic, and social sustainability [22,39]. Good sustainable management of agriculture allows for equitable, viable, and appropriate services [17,40].

Eco-efficiency is a practical concept to assess the sustainability of economic behavior and obtains the maximum development of economic benefits with the minimum impact on the ecological environment [20,23]. The definition adopted by the World Business WBCSD in 1992 advocated eco-efficiency to assess and promote sustainable production in business [5,26]. The concept was subsequently extended and developed in agriculture, industry, and other related fields. Among them, agriculture is one of the main sectors where the concept of eco-efficiency has been applied [4,21]. The measurement of AEE is a quantification of the relationship between environmental impacts and agricultural production [17,18], as well as a powerful support for the development of sustainable policies in agriculture. Numerous researchers have conducted multi-method and multi-functional analytical studies around AEE and proposed multi-perspective program measures based on the constraints of regional agricultural sustainable development. Yang performed an AEE analysis in China based on carbon emission indicators to quantify the impact of carbon emission reduction on agricultural productivity [14,19,20]. Coluccia analyzed AEE in Italy based on surface source pollution indicators to propose policy recommendations for the sustainability of agricultural production [8]. Grassauer completed data envelopment analysis (DEA) to explore why the decrease in the eco-efficiency of Austrian farms provoked the increase in human demand for meat protein [29]; Maia investigated the environmental and economic constraints of eco-efficiency around Monterovo irrigation in Portugal assessed by Life Cycle Assessment (LCA) based on the water value chain context and suggested resource-saving and environment-friendly as the focus of efficiency improvement [5]. Additionally, regional AEE has been researched from multiple perspectives, such as institutional change, technological heterogeneity, and crop diversity. Stępień revealed the relationship between eco-efficiency and institutional variables based on New Institutional Economics (NIE), suggesting that a small farm payment system under the policy increased residential consumption and reduced eco-efficiency [22]. Han adopted the Meta-Hybrid-U approach to characterize the distribution of eco-efficiency under technical heterogeneity and unveiled that AEE is multiply positively distributed with both of them under the joint effect of management and technology [9].

Current research illuminates the characteristic status of AEE in different contexts and regions and further analyzes the reasons for its changes. Nevertheless, the AEE measurement system requires further improvement. Moreover, the driving mechanism and related policy recommendations need further refinement. There are two innovations in the research. First, agroecosystems continue to provide substantial direct or indirect benefits to humans, whereas they have not been adequately represented in previous studies of agroecological efficiency. Hence, the value of agroecosystem services is incorporated as part of the desired output into the evaluation index system in this paper. Second, innovation is reflected in the research perspective. Based on the overall AEE of the region, the changing trend of the center of gravity in the context of “double carbon” is investigated, and policy advice is added on the weak links of regional agricultural development in accordance with its displacement characteristics and constraints.

## 3. Materials and Methods

### 3.1. Study Area

The YRD (Yangtze River Delta) region is located in the downstream area of the Yangtze River in China, where the Yangtze River, the Yellow Sea, and the coastal area meet. The total area is 225,000 square kilometers. As illustrated in Figure 1, the area is flat, with abundant rainfall, and the terrain is mainly alluvial plains formed by the Yangtze River entering the sea, with dense rivers and lakes, mainly including the famous Taihu Lake, Hongze Lake, and Gaoyou Lake. There are some hills and low mountains with relatively low elevations scattered in the YRD. In 2020, the total GDP of the YRD region was 2,447,318 billion, accounting for 24% of the national GDP (1,015,986.2 billion yuan). In the same year, the value added of agriculture in the YRD region was over 12% of the total agricultural economy of China. With a dense population, developed agriculture, and abundant products, it is one of the regions with the most active economic development, the highest degree of openness, and the strongest innovation capacity in China.

### 3.2. AEE Indicator System

#### 3.2.1. Indicator Design and Measurement

AEE integrally reflects the integrated development relationship between the three elements of agricultural economic growth, resource conservation, and environmental protection. In the process of agricultural production, production factor inputs yield corresponding desired outputs, accompanied by the production of various pollutants [18]. Desired outputs are beneficial outputs, whose production enhances human well-being, in which gross agricultural product is the total outcome of agriculture, forestry, livestock, and fishery, and ecosystem services are the total value of services provided to humans by the agricultural ecosystem, both of which more comprehensively represent the overall value output of agriculture [10,11]. Non-desired outputs are harmful production outcomes, the production of which can have negative impacts on human life and arise mainly from excessive or inefficient use of certain production factor inputs [9].

Based on the aforementioned and related literature [5,41,42], six input factors are selected in this paper, including land, livestock, electricity, labor, mechanical power, and agricultural materials (data were collected from the Statistical Yearbook in cities, Land Use Status Survey Report, and the Bulletin of Key Data of Land Survey). The land is measured as “crop area sown”, which is more accurate than “arable land” in measuring the actual utilization of land; livestock, electricity, mechanical power, and agricultural materials are used as part of the capital inputs; labor is directly used as the number of people employed in agriculture. All of these factors produce significant amounts of greenhouse gases. There are two desired output elements of agriculture. Among them, the total value of agricultural production is derived from the China Rural Statistical Yearbook, and the value of agricultural ecosystem services is calculated from Equations (5) and (6). The non-desired output elements are carbon emissions [11,20]. The description of their indicators is detailed in Table 1.

AEE is measured using the Super-SBM model, which is an optimized upgrade of the DEA model. DEA combines inputs and outputs with custom weighting coefficients to obtain corresponding scores that indicate the relative efficiency of decision-making units (DMUs) [30] and is commonly used to measure AEE. It is difficult for the traditional DEA model to distinguish the efficiency differences in multiple decision units with the same maximum efficiency value of 1, and the problem of overestimating agroecological efficiency occurs since its principle produces slack variables that are not zero [5]. The SBM model proposed by Tone et al. is a supplement to the DEA model, which effectively solves the input–output slack phenomenon and the juxtaposition problem of ranking [18]. The super-SBM model, designed by Zhou et al., contributed to a further improvement of the SBM model [44], which combines the advantages of the super-efficient DEA model and SBM model, and incorporates non-desired outputs into the model to achieve the effect of optimizing eco-efficiency. It is widely used to measure eco-efficiency in different sectors and is the mainstream model for determining AEE, expressed as [30]:(1)$$p=min\frac{\frac{1}{m}\sum_{i=1}^{m}\left(\frac{\bar{x}}{x_{ik}}\right)}{\frac{1}{s_{1}+s_{2}}\left(\sum_{s=1}^{s_{1}}\frac{{\bar{y}}^{g}}{y_{sk}{}^{g}}+\sum_{q=1}^{s_{2}}\frac{{\bar{y}}^{b}}{y_{qk}{}^{b}}\right)},$$
(2)$$\left\{\begin{array}{c} \bar{x}\geq \sum_{j=1,\;\;\neq k}^{n}x_{ij}\lambda_{j};\;{\bar{y}}^{g}\leq \sum_{j=1,\;\neq k}^{n}y_{sj}{}^{g}\lambda_{j};\;{\bar{y}}^{b}\geq \sum_{j=1,\;\neq k}^{n}y_{qj}{}^{b}\lambda_{j};\\ \bar{x}\geq x_{k};\;{\bar{y}}^{g}\leq y_{k}{}^{g};\;{\bar{y}}^{b}\geq y_{k}{}^{b};\\ \lambda_{j}\geq 0,\;i=1,\;2,\;\ldots ,m;j=1,\;2,\;\ldots ,n;\\ j\neq 0,\;s=1,\;2,\;\ldots ,s_{1};q=1,\;2,\;\ldots ,s_{2}; \end{array}\right.,$$
where *p* represents the efficiency evaluation value; *x*, $y^{g}$, and $y^{b}$ denote the values of the input, desired output, and non-desired output indicators, respectively; *m* indicates the number of input indicators; $s_{1}$ and $s_{2}$ stand for the number of desired output indicators and non-desired output indicators, respectively; *n* is the number of DMUs (decision making units), *n* = 16 (the number of cities in the study area), and each DMU consists of m inputs, $s_{1}$ desired outputs, and $s_{2}$ non-desired outputs; *λ* signifies the weight of the corresponding input or output element.

#### 3.2.2. Agricultural Carbon Emission Estimation Methods

Agricultural production is also one of the main sources of carbon emissions. A large amount of carbon emissions causes non-normal changes in climate and is an undesired output of agricultural production [20]. Carbon emissions of pollutants as the corresponding factor inputs and outputs are vital components of measuring regional AEE. Based on relevant literature [33,45] and regional industrial development, three types of carbon sources are selected in the paper: land management, livestock farming, and crop cultivation. Carbon emissions from land management refer to carbon emissions directly or indirectly caused by human actions in the process of agricultural production and its activities, including agricultural fertilizers, pesticides, agricultural diesel, agricultural film, agricultural machinery power, agricultural irrigation, and tillage, for a total of seven carbon sources; carbon emissions from livestock farming refer to CO_2_, CH_4_, and N_2_O emissions produced by pigs, cattle, and other livestock during their natural growth cycle through gastrointestinal fermentation and manure management systems; carbon emissions from crop cultivation are greenhouse gas emissions caused by the destruction of the surface layer for agricultural soils during the natural growth of crops, with N_2_O being the most prominent [46,47,48]. According to Tian Yun et al. [49,50] and (IPCC) [51] Fourth Assessment Report research results, a more localizable comprehensive evaluation method of agricultural carbon emission coefficients is used to calculate carbon emissions amount [49]:(3)$$E=\sum_{i=1}^{n}E_{i},$$
(4)$$E_{i}=\sum_{k=1}^{n}T_{ki}\alpha_{ki},$$
where *E* denotes the total agricultural carbon emissions; $E_{i}$ represents the carbon emissions from *i* carbon source; $T_{ki}$ indicates the amount of *k* carbon sources; $\alpha_{ki}$ signifies the carbon emission coefficient of *k* carbon source; and $\alpha_{ki}$ is the carbon emission coefficient of carbon sources of *k* minor categories and the emission coefficient combined with the study of IPCC. The emission coefficients are detailed in Table 2.

#### 3.2.3. Methods for Estimating the Value of Farmland Ecosystem Services

Ecosystem services refer to all the benefits that can be obtained directly or indirectly from agroecosystems in human activities; agroecosystem services provide a large number of benefits to humans [53] and are the desired output component of agricultural production. Scholars account for the value of services with the InVEST model [54,55], equivalent price transfer method [42,53], and the value equivalence factor method [53,56,57], among which the value equivalent method uses standard equivalence factors to calculate, which can reduce human factor interference [58]. Therefore, this article adopts the base equivalence table of the value of ecosystem service function per unit area of Xie et al. 2011 [56], and optimizes the value of agroecosystem services in 26 municipalities in the YRD region by combining the data of the year-end arable land area and agricultural output value, which measures the value from four aspects, namely, supply service, regulation service, support service, and cultural service, with the formula [11]:(5)$$EVA_{i}=\sum_{k=1}^{n}Z_{ki}S_{i}F\frac{W_{1}}{W_{2}},$$
(6)$$EVA=\sum_{i=1}^{n}EVA_{i},$$
where $EVA_{i}$ represents the value of ecosystem services in i municipality; $S_{i}$ stands for the area of farmland in i city; *F* refers to the value equivalent factor; Zn denotes the base equivalent of ecosystem services of n category; $Z_{ki}$ indicates the value equivalent of class *K* ecosystem services in i city; $W_{1}$ signifies the total agricultural output value in the YRD region; $W_{2}$ expresses the total agricultural output value in the national region; and EVA reflects the value of agroecosystem services in the YRD region.

### 3.3. Spatial Autocorrelation Analysis

Spatial autocorrelation is employed to test whether the observations of a unit are spatially correlated with those of its neighbors. It is divided into global autocorrelation analysis and local autocorrelation analysis, measuring the spatial association of the region as a whole and locally [59]. The article obtained the spatial distribution characteristics of AEE in the YRD region by importing the data into ArcGIS software (https://www.esri.com/en-us/arcgis/about-arcgis/overview (accessed on 6 March 2023))and classifying the relevant data hierarchically based on the natural discontinuity method. Spatial autocorrelation is the representation of the Moran index [10]. The global and local spatial characteristics of ecological benefits in the YRD region are explored using the Moran index to analyze the spatial and temporal characteristics of recognition benefits and reveal development law. The formula [60] is:(7)$$I_{a}=\frac{\sum_{i=1}^{n}\sum_{j=1}^{n}\left(x_{i}-\bar{x}\right)\left(x_{j}-\bar{x}\right)}{S^{2}\sum_{i=1}^{n}\sum_{j=1}^{n}w_{ij}},$$
(8)$$Z=\frac{1-E\left(I\right)}{\sqrt{Var\left(I\right)}},$$
(9)$$I_{i}=\frac{\sum_{j=1}^{n}w_{ij}x_{j}-\bar{x}\sum_{j=1}^{n}w_{ij}}{S\sqrt{\frac{n\sqrt{\sum_{j=1}^{n}w_{ij}{}^{2}-{\left(\sum_{j=1}^{n}w_{ij}\right)}^{2}}}{n-1}}},$$
where $I_{a}$ denotes the global Moran index value; *n* represents the total number of study units; $W_{ij}$ refers to the spatial weights; $x_{i}$ and $x_{j}$ indicate the observed values in region i and region *j*, respectively; *Z* reflects the significance of the global Moran’s *I*; $E\left(I\right)$ and $Var\left(I\right)$ are the expected value and covariance of the global Moran’s *I*, respectively; $I_{i}$ signifies the local Moran’s *I* values; and *S* stands for the standard deviation. $I_{i}$ > 0 indicates a strong positive spatial autocorrelation between region i and the neighboring regions, which is a high-value aggregation area, and $I_{i}$ < 0 implies a low-value aggregation area.

### 3.4. Identification of Influencing Factors Based on Regression Analysis

#### 3.4.1. Variable Description

The factors influencing AEE were first assessed regarding the direction of influence by relevant theories, and the results were verified by the regression analysis. The current situation of agricultural and rural development in the YRD reveals its characteristics and the collectability of data [22], as shown in Table 3. Nine factors, such as Ami, Ais, Url, and Ccs, were selected to analyze the impact on AEE [18]. Among them, Eir is the area of arable land being effectively irrigated, Url is the percentage of the urban population, and Ami is the power generated by the use of medium and large machinery. They are all closely related to agricultural modernization and agricultural high technology with a positive orientation to AEE [61,62]. Additionally, Ecv is the amount of electricity used by agriculture and its activities, Ais is the share of the primary sector in the gross product, and Ccs is the share of crop sown area. The three exist to yield desired outputs and non-desired outputs of insignificant strength and have an unclear direction of influence on AEE [18,19]. Scl is the area of arable land occupied per capita, in which the more the extensive crop planting area, the more the non-desired output, leading to an increase in AEE [63]. The pca income ratio is the ratio of urban to rural residents’ per capita disposable incomes. High income is accompanied by an expansion of factor inputs, and there is a situation of excessive emissions, which hurts AEE [18]. Afi is the discounted amount of agricultural fertilizer applied per hectare of arable land. Excessive use of fertilizer can induce severe waste of resources, soil pollution, and lessened AEE [21].

#### 3.4.2. Tobit Model

Regression models are widely used in the investigation of variable influencing factors, among which the most common regression methods are Ordinary Least Squares [18,64], Spatial Durbin model [65], Tobit [10,20], and GWR [66]. The Tobit model is a model in which the dependent variable takes values in a continuous but restricted situation by some factors, and it applies to the case in which the dependent variable y has a zero value while other values are positive and continuous [20]. Compared with other models, the Tobit regression model can effectively solve the above problem of continuous dependent variables with truncated values [10], reduce the bias of the estimated parameters, and improve the accuracy. With reference to the dynamic characteristics of AEE, geographical characteristics, and related literature [10,18,20], nine influencing factors related to agricultural production were selected to explore the restricted situation of influencing factors, with AEE values as the restricted variables, through Tobit panel random effects. The formula [10] is:(10)$$\left\{\begin{array}{c} y=\alpha +\beta x+\sigma ,\;y>0\\ 0,\;y\leq 0 \end{array}\right.,$$
where *y* denotes the vector of dependent variables, which is agroecological efficiency in this paper. α is the vector of intercept terms; β represents the vector of regression parameters; x is the vector of independent variables, including Eir, Url, Ais, Ccs, Scl, Ami, Ecv, Pca, Afi; σ refers to the random error term.

#### 3.4.3. Geographically Weighted Regression Model

The Geographically Weighted Regression (GWR) model, which was first proposed by Brunsdon [18], is a local linear regression model used to study the spatial heterogeneity characteristics of variables. Compared with other models, GWR introduces spatial characteristics into the regression parameters, breaking the non-smoothness and dependence of the original spatial data [66]. The method obtains correlation results by importing sample point vector data in GIS. Its main application is regression analysis in the presence of autocorrelation scenarios. It is expressed as [66]:(11)$$\ln g_{i}=w_{0}\left(m_{0},n_{0}\right)+\sum_{i=1}^{k}x_{ij}\left(m_{0},n_{0}\right)\beta_{i}+\varepsilon_{i\;}\left(i=1,2,3,\cdots ,n\right),$$
where $\ln g_{i}$ is the logarithm of the ith eco-efficiency; $\left(m_{0},n_{0}\right)$ is the central geographical coordinate of eco-efficiency; $\left(m_{0},n_{0}\right)\beta_{i}$ is the regression coefficient of the *i*th eco-efficiency; $x_{ij}$ is the influencing factor of AEE; and $\varepsilon_{i}$ is the random error term.

## 4. Results

### 4.1. Temporal and Spatial Variation Characteristics of AEE

The YRD’s agricultural economy is more developed than in other parts of China, and it is an earlier area in China’s agricultural reform and opening up. Its agricultural ecological benefits are at a higher level in China. In the past 21 years, the range of AEE in the whole region was 0.75–1.01, and the average annual AEE was 0.83. The regional variation trend demonstrated an overall increase, as presented in Figure 2. From 2000 to 2003, the overall AEE in the YRD decreased by 0.03 since the natural disasters occurred in China during the previous period and the national focus on the development of secondary and tertiary industries neglected the development of related agriculture [5,67]. After 2003, especially from 2007 to 2020, the AEE of the YRD region exhibited a linear increase, from 0.75 to 1.01, with an overall increase of 0.26. This was caused by the national emphasis on agroecology and production construction. After 2003, the state introduced a series of tax-free subsidies for agriculture [10], boosting the investment of funds in the agricultural industry. Additionally, the agroecological efficiency of the YRD region displayed temporal heterogeneity while there were more significant geographical differences. Shanghai, the most economically and technologically developed region in the entire YRD, with a flat territory and advanced agricultural technology, has a multi-year average AEE value of 0.98, which is the highest efficiency value in the region. In 2000, the efficiency value was significantly higher than that of other regions, and subsequently illustrated an up-and-down trend of AEE changes, with a low level of increase and little room for improvement. The AEE value in Jiangsu was the second highest after Shanghai, and its multi-year average value was 0.87. The trend was mainly U-shaped, with a decreasing trend from 2000 to 2005 and an exponential increase from 2007 to 2018, followed by a short decline in 2019 and a return in 2020. The levels of AEE in the Zhejiang and Anhui regions were lower than the overall level of AEE in the YRD region, and the magnitude of their relative efficiency values kept changing over time. From 2000 to 2006, the level of AEE in the Zhejiang region was higher than that in the Anhui region, while Anhui surpassed Zhejiang in 2006–2014, and Zhejiang surpassed Anhui and made the gap widen gradually after 2015. The changes and distribution characteristics of AEE in the YRD region suggested a strong correlation between the state of agroecological development and regional economic and scientific development. Anhui and Zhejiang regions adopt more traditional rough agricultural production. Nevertheless, their AEE improvement space is further expanded with the inclination of policies and funds, and with the promotion of agricultural structure reform.

Concerning the regional distribution, the agroecological efficiency in the YRD region increased in a gradient over 21 years, with a clear differentiation in spatial rank and significant differentiation characteristics in spatial and temporal evolution. According to relevant researchers [68], the AEE values are divided into five classes: low efficiency (0.5 ≥ e > 0.7), little low efficiency (0.7 ≥ e > 0.8), medium efficiency (0.8 ≥ e > 0.9), little high efficiency (0.9 ≥ e > 1), and high efficiency (e ≥ 1), which present the characteristics of spatial distribution classes of AEE in different spatial and temporal states by municipal geography, as well as the characteristics of AEE changes in each region during the period of 2000–2020 by the slope. As illustrated in Figure 3, the regionalization of AEE levels in the YRD region is significant. The areas with high levels of multi-year average efficiency values in the YRD region are mainly distributed in eight cities in the northeast, including Yancheng, Taizhou, and Yangzhou. The areas with low levels are distributed in 18 cities in the Anhui and Zhejiang regions, including Chuzhou, Hefei, and Hangzhou in the southwest. This is majorly associated with the flat topography, excellent arable land resources, developed agricultural economy and technology, as well as local government support for relevant agroecological green development. Additionally, the polarization of AEE in the YRD region has weakened, and the regional balance has increased, but the regional grading of the evolution of efficiency values was significant. The area of the optimized AEE in the four time periods from 2000 to 2020 has been expanding over time; the area share of low-efficiency areas has shrunk from 69% to 7.59%, and the number of medium- and high-efficiency cities has increased from 8 in 2000–2005 to 24 in 2016–2020, covering almost the entire region. The area of the region exhibiting an increase in AEE value expanded from 34.53% in 2000–2005 to 92.31% in 2016–2020. Among them, the regions with a significant increase in AEE values were primarily observed in the southwest (Anhui and Zhejiang regions). With the deepening development of agricultural reform, the agricultural industrial structure of the YRD region has been optimized, the regional AEE has achieved higher results, and the balanced regional development has been improved.

### 4.2. Spatial Correlation Characteristics of Agroecological Efficiency

A certain degree of spatial autocorrelation exists in agroecological efficiency, and information from different regions affects the presentation of results from neighboring regions [11]. The YRD region generally presents a high degree of aggregation but different degrees of aggregation in different time periods, and its spatial dependence has temporal heterogeneity. As revealed by taking the average value of AEE for 21 years and eliminating the influence of multiple factors, Moran’s I value for the YRD region was 0.51, and the Z value was 4.39 (*p* < 0.01). There was a significant spatial autocorrelation overall, and municipalities with similar levels of AEE demonstrated aggregation or dependence distribution in space. With respect to time periods, the global autocorrelation of agroecological efficiency in the YRD region over four time periods was a “clustering–random” process, with the highest Moran’s I index of 0.36 in the 2006–2010 and 2011–2015 periods, both of which passed 1% (Table 4). The highest aggregation of AEE in these two periods is closely correlated with the relevance of intra-regional policies, cross-regional flow of production materials, exchange and learning of agricultural technology, and trade flow of agricultural products, especially the timeliness of policies.

The global spatial autocorrelation suggests that the YRD region presents high aggregation for many years. Next, the aggregation status of the YRD region is spatially analyzed through local autocorrelation. With GIS, the units involved in spatial correlation were classified into four categories: high–high, low–high, high–low, and low–low. The AEE among the high–high aggregation areas in the YRD has the characteristics of mutual promotion and joint improvement, while the AEE of the low–low aggregation areas has the characteristics of an equally low level, owing to the constraints of natural and social conditions. Thus, high–high agglomerations and low–low agglomerations are classified as hot spot areas and cold spot areas, respectively [12]. It is unveiled that the spatial dependence of agroecological efficiency in the YRD region is characterized by polarization, with Jiangsu and Shanghai in the northeast as hot spot areas and Anhui and Zhejiang in the southwest as cold spot areas. Because of the good natural conditions, convenient transportation, and developed socio-economics in Jiangsu and Shanghai, the regions during 21 years are frequently accessed and exhibit a high–high clustering area in general, comprising 3 cities, Taizhou, Nantong, Shanghai,, accounting for 11.54% of the number of administrative regions. Due to its topography and other natural conditions, the overall low–low aggregation area is mainly located in the southwest of Anhui and Zhejiang, including Xuancheng, Shaoxing, Hangzhou, and Taizhou, accounting for 15.38% of the number of administrative regions. Additionally, there is temporal variability in the AEE aggregation areas in the YRD region, and the inter-regional correlation is gradually weakening. With the continuous passage of the time period, the hot spot area gradually shifted to Yangzhou, and the scope of the clustering area gradually shrank from the original three cities to two cities (Figure 4). There was high and low clustering in the 1st, 2nd, and 3rd stages. Its distribution range was in the northwestern part of the Anhui region and the southwestern part of the Zhejiang region. The cold spot area gradually moved to the southwest of Anhui, and the clustering range shrank to the same extent. It increased from two in stage one to four in stage two, and then gradually decreased to one in stage four.

### 4.3. Analysis of Factors Influencing AEE

Different human activities have different degrees and directions of influence on AEE. The influence of each factor on AEE in the YRD was explored through the regression analysis of Tobit and used GWR for the comparison test. After the test of *p* < 0.005, the results of the two studies were more consistent, with no substantial changes and in the same direction of influence as the previous forecast factors. Url, Ccs, Ais, Scl, and Ami had positive effects on AEE, ranging from 0.04 to 0.49 and 0.04 to 0.39. The largest effects were on Url (0.49), Ais (0.22), and Ccs (0.21) (Table 5). For each 1% increase in urbanization, the regional AEE increased by 0.49%. In the regions with more developed urbanization, their agricultural mechanization is high, their agricultural output is high, their pollution is low, and the input means of production can obtain a high proportion of output, contributing to the increased value of AEE. Secondly, a good structure of the agricultural industry and crop cultivation can optimize input costs and obtain more desired outputs with lower ecological impacts. Ecv (−0.04), Pca (−0.05), and Afi (−0.17) harm AEE in the YRD region, of which the largest impact is from Afi, and regional AEE decreases by 0.17% with a 1% increase in agricultural fertilizer use intensity. The reason is that the more fertilizers are used in the agricultural production process, the more pollution is produced in the natural environment, such as land and water bodies, so as to lessen the regional AEE. Therefore, the level of urbanization, the structure of the agricultural industry, and the use of fertilizers should be reasonably planned and designed to formulate agricultural green policy measures.

## 5. Discussion

### 5.1. Analysis of the Evolution of the Trend of the Center of Gravity of AEE for Low-Carbon Development

China is a largely agricultural country and currently has been facing the headaches of declining arable land quality, agricultural pollution, degradation of agricultural ecosystems, and agricultural carbon emissions [69]. To grapple with the related complications, the state proposed the “low carbon” and “double carbon” policies and actively promote the green low-carbon development of agriculture. In this context, the center of gravity of AEE in the YRD region has slightly shifted to the southwest, while it is generally stable and mainly located in the border area of three provinces, because the levels of AEE and growth in the three provinces are roughly the same. The slight shift is provoked by the low starting point of AEE in Zhejiang and Anhui in the southwest, and has been significantly improved with the investment of policies, technology, and capital, presenting large room for growth. As illustrated in Figure 5, the trajectory of AEE in the YRD region is divided into two time periods: 2000–2009 and 2010–2020. In the first 10 years, the center of gravity moves mainly within Jiangsu province, with a fluctuating and zigzagging movement, and the migration path is mainly in the northeast–southwest direction. In the second 11 years, the center of gravity moves from Jiangsu province to Anhui province, the migration range is larger, and the migration path is primarily in the north–south direction.

The AEE moved in the direction of favoring Anhui Province from 2000 to 2009. At the beginning of the 21st century, the state formulated a series of plans for the construction of the central and western regions and boosted investment in infrastructure construction, environmental improvement, and technological progress. In 2007, the Chinese government further proposed the concept of “ecological civilization” and shifted the government’s orientation of agriculture to modern agriculture with ecological agriculture. Anhui Province, with the development of its economy, has invested more money and technology in the green development of agriculture. While the AEE of each province has increased, Anhui Province has increased more significantly, making the AEE more than Zhejiang Province in the period after 2006. Meanwhile, the center of gravity of AEE has shifted northeast. From 2010 to 2020, Zhejiang’s agricultural economy developed at a high speed, and Zhejiang proposed efficient eco-agriculture with the purpose of optimizing the internal structure of agriculture and promoting mechanized production and low-carbon agriculture steadily. This contributed to 3% of China’s agricultural value added with 1.3% of China’s arable land in 2017, significantly increasing Zhejiang’s AEE during this decade, which surpassed Anhui Province after 2014 and shifted the center of gravity of AEE southward.

### 5.2. AEE Optimization Analysis

The development of AEE in the YRD is characterized by regional incoherence, variability, spatial dependence, and multi-factor drive. The multi-year average AEE in the YRD from 2000 to 2020 is high in the northeast and low in the southwest, with significant geographical differences (Figure 6). This is associated with the natural conditions of the provinces and the degree of economic and technological development [70]. Moreover, the northeastern regions of the Jiangsu, Shanghai, and Zhejiang regions are economically developed, and a good economic foundation can effectively support the development of ecological agriculture and related agricultural carbon reduction [71]. To wrestle with the uncoordinated AEE, government regulation can be used to increase policy support and financial investment, while peer-to-peer financial and technological support and cooperation from high-efficiency regions to low-efficiency regions can be applied to shrink inter-regional differences in AEE and achieve balanced development [14,16]. Meanwhile, each region should implement different agricultural policies following realistic conditions. Considering that the northeastern region of the YRD has vast plains and is the main grain-producing area, farmland construction can be planned to accelerate the process of achieving full mechanization coverage and boost the desired output. The southwestern part, which is mainly mountainous and hilly, should reduce carbon emissions and surface pollution and perform refined agriculture, such as planting forest crops with a high balance of ecological, carbon sink, and socio-economic values.

There is a regional correlation of AEE between the YRD regions, with high aggregation in the northeast and low aggregation in the southwest. This is induced by the mobility and dependence of inter-regional agricultural series (production materials, agricultural policies, agricultural technologies, and agricultural products) [12]. The flat terrain, convenient transportation, and high information mobility in the northeastern region allow its interregional agroecological efficiency to be mutually reinforced and exert a positive regional effect. The southwest is mostly mountainous and hilly, with unchanging traffic and slow information flow, leading to a low level of inter-regional agroecological efficiency and showing correlation. Therefore, the radiation range of the high–high aggregation region should be increased, and further extension should be performed for the surrounding cities (consisting of policies, technology, and production factors). Low–low gathering areas should optimize the construction of transportation and promote the interconnection and sharing of roads, water, electricity, networks, gas, energy, and other infrastructure, urban and rural. The joint mutual assistance of regional agricultural development should be strengthened, and an electronic data online exchange platform should be established, so as to contribute to the exchange and transmission of agricultural science and technology knowledge, agricultural trade, and regional government policy information. Meanwhile, the rational flow of production should be reinforced, and a new pattern of mutual collaboration and common progress of agricultural green development should be further formed.

The multiple input elements of AEE are influenced by various factors. AEE in the YRD region is majorly impacted by three aspects: socio-economic level, agricultural industry structure, and production material inputs. The Url in the paper has a crucial positive contribution to AEE, reflecting that modern agriculture in the region is still the main agricultural development mode for improving agroecological and economic values at present. Nevertheless, the small influence of the mechanization level validates that the current mechanization level in the region still needs to be improved, and its role on AEE in the YRD region possesses more room for play. Therefore, it is urgent to emphasize the degree of regional machinery use, increase the level of scientific and technological investment in rural production, improve the level of labor quality, and coordinate the urbanization process while continuing to promote the optimization of agricultural production, so as to boost the inflow of rural capital and develop a sufficient number of laborers engaged in agricultural production [21]. Ais and Ccs are also the main influencing factors of regional AEE. Optimizing for market orientation, adjusting the structure of products in each sub-sector of agriculture, forestry, animal husbandry, and fishery, as well as the input structure of production factors, bringing into play the agricultural industry with the advantages of regional conditions, and improving the efficiency of energy use are vital directions for measures that aim to improve the regional agricultural economic efficiency, environmental efficiency, and reduce carbon emissions. Concurrently, the intensity of agricultural production imposed by fertilizers and other inputs significantly cuts the level of AEE in the region. In the excessive and inefficient use of chemical fertilizers, a large amount of surface pollution and carbon emissions can increase the non-desired output. Hence, agricultural pollution could be curtailed by introducing clean agriculture models, increasing the use of organic fertilizers, performing soil testing and fertilization, and employing plant protection drone technology.

### 5.3. Limitations and Prosperity

There is variability in AEE due to the diverse methods and different agricultural indicators adopted by different researchers [15,20]. Concerning the construction of non-desired output indicators of agroecological efficiency, Yang et al. [20,21] supplement the output of land and water pollution caused by chemical fertilizers and pesticides with carbon emission and surface source pollution as non-desired outputs. Compared with the former, the input indicators are improved in this paper by adding livestock and electricity indicators, and desired output indicators include the value of agricultural ecosystem services, which enhances the comprehensiveness of the input and the desired output indicators. However, in the selection of non-desired output indicators, ignoring the agricultural surface source pollution indicators results in some bias in the AEE. In the study of the influencing factors of regional agroecological efficiency, variability exists in the influencing factors attributed to the selection of indicators and regional restrictions [72]. Liu et al. used rainfall, farmers’ education levels, and agricultural market stability as the limiting factors of their regional agroecological efficiency [8,18] under the consideration of the effects of climate change, rural population quality level, and market changes on agroecological efficiency. In this paper, the impact of factors such as socio-economic development, agricultural technology, agricultural structure, and agricultural fertilizer application intensity during the low carbon development on AEE is analyzed. Nonetheless, there is little effective consideration of the relevant contents of the above researchers. Therefore, in future studies, pollutant outputs of factor inputs, such as surface sources, are increased regarding indicator selection, and more optimal methods are selected in indicator calculation to further construct a more complete indicator system and obtain more accurate indicator results. With respect to future constraints, the influencing factors such as meteorological changes, agricultural labor force transformation, policy changes, and other influencing factors, as well as sub-regional exploration of AEE influencing mechanisms, should be further increased to explore the attribution of AEE changes more comprehensively and deeply, contributing to laying a reference foundation for precise management of agricultural regionalization and promoting the optimal use of agricultural resources and green and coordinated development. Additionally, the mathematical models used in the paper will be further optimized in the course of subsequent research. Specifically, the results of the value of agroecosystem services estimated by the value-equivalent method in this paper will be revised using an ecological remote sensing model; the Spatial Durbin model will be employed to determine the influencing factors and reveal their spatial spillover effects.

## 6. Conclusions

This study adopts a suitable agricultural production index method and an optimized agricultural production index structure. The development status, regional characteristics, and driving mechanisms of AEE in the YRD region are analyzed through the model, and the current situation of agricultural green production in each region is explored from the perspective of efficiency center of gravity migration in a low-carbon context. The conclusions are drawn as follows:(1)The AEE of the YRD region is at a high level in China, averaging around 0.83 throughout the year. In terms of the temporal trend, the YRD region as a whole presents a U-shaped movement. It decreased slightly from 2000 to 2003 and increased exponentially after 2003. The efficiency value increased by 0.05 compared with the overall in 2000, with the largest increase from 2007 to 2020. Regarding spatial evolution, the high-level area of AEE in the YRD region spread from the northeastern Jiangsu and Shanghai regions to the southwestern Anhui and Zhejiang regions in a stepwise manner, with significant grade differences and significant polarization in the degree of increase: high in the southwest and low in the northeast. The overall high-efficiency area was the radiating cities centered on Yancheng and Shanghai, while the low-efficiency area was the surrounding cities centered on Wuzhou and Jinhua.(2)AEE in the YRD region generally had significant spatial correlation and dependence while exhibiting spatial and temporal heterogeneity. The 21-year average efficiency values suggested that the regional space has strong autocorrelation, among which Taizhou, Nantong, and Shanghai were clusters with high agricultural efficiency values, and Maanshan, Xuancheng, Hangzhou, Shaoxing, and Taizhou were clusters with low AEE values. With respect to time series, AEE in the YRD region displayed a “clustering–random” trend, and inter-regional correlations were weakened.(3)Except for the insignificant effect of Eir, the effects of socio-economic, agricultural production structure, and agricultural production factors were all significant at the 0.01 level in the YRD. The impact of the influencing factors on agricultural ecological efficiency is in the following order: Url, Ais, Ccs, Scl, Ami, Ecv, Pca, and Afi. Among them, urbanization level, agro-industrial structure, and crop structure were the main positive influencing factors of AEE in the YRD, and fertilizer use intensity was the main negative influencing factor.(4)The center of gravity of AEE in the YRD region under the influence of agricultural green and low-carbon policies slightly shifted to the southwest. The center of gravity shifted mainly to the northwest from 2000 to 2009. Furthermore, the center of gravity shifted mainly to the south from 2010 to 2020, and was stable in the border area of the three provinces.

The results unveil significant regional disparities in the development of AEE in the YRD, as well as the inconsistent focus of agricultural policies and agricultural science technology in each region. Given the complications of insufficient motivation for subsequent agricultural green development in the northwest and the need to improve the level of machinery in each region, local governments and agricultural organizations should plan innovations in agricultural green policies and agricultural science technology programs while strengthening transportation links between regions to improve the circulation of agricultural production factors.

## Figures and Tables

**Figure 1 ijerph-20-04786-f001:**
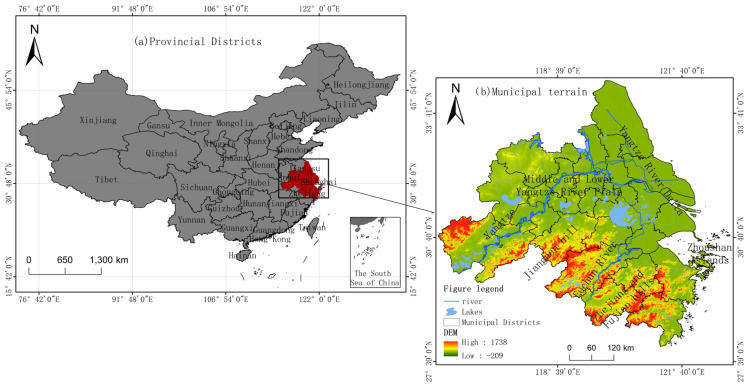
Overview of the study area.

**Figure 2 ijerph-20-04786-f002:**
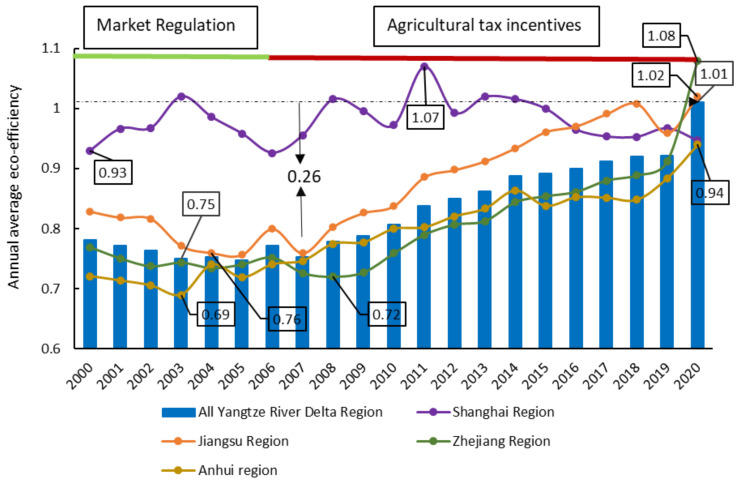
Trends in the time variation of AEE in the YRD region from 2000 to 2020.

**Figure 3 ijerph-20-04786-f003:**
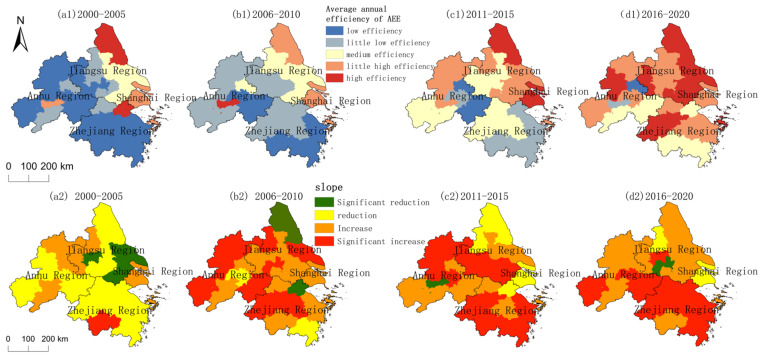
Spatial distribution of annual average AEE and slope changes in the YRD at four time periods.

**Figure 4 ijerph-20-04786-f004:**
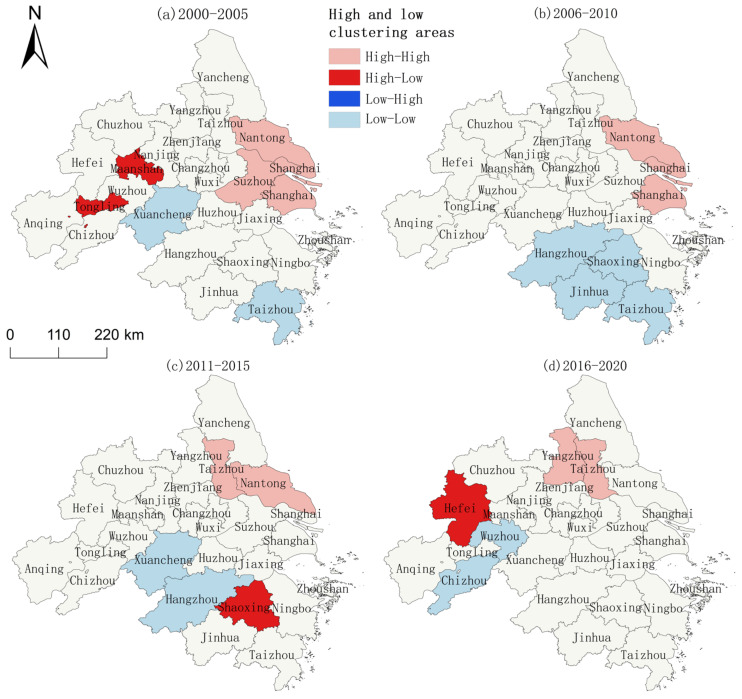
Spatial distribution of AEE clusters in the YRD region in four time periods.

**Figure 5 ijerph-20-04786-f005:**
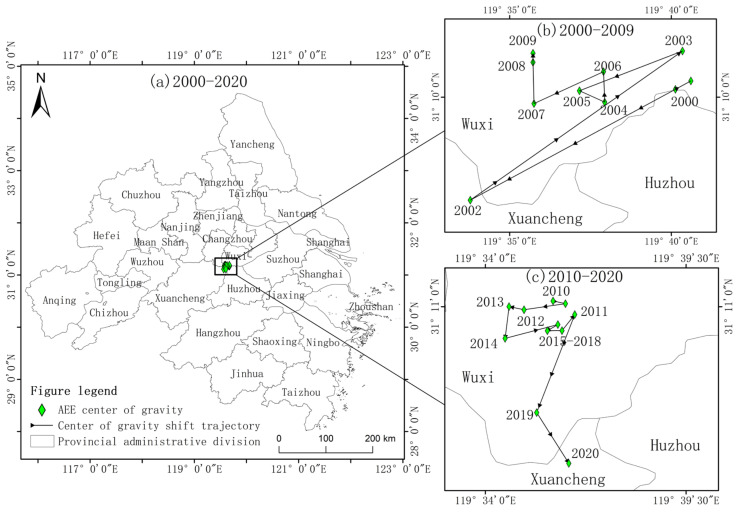
The center of gravity shift of AEE in the YRD from 2000 to 2020.

**Figure 6 ijerph-20-04786-f006:**
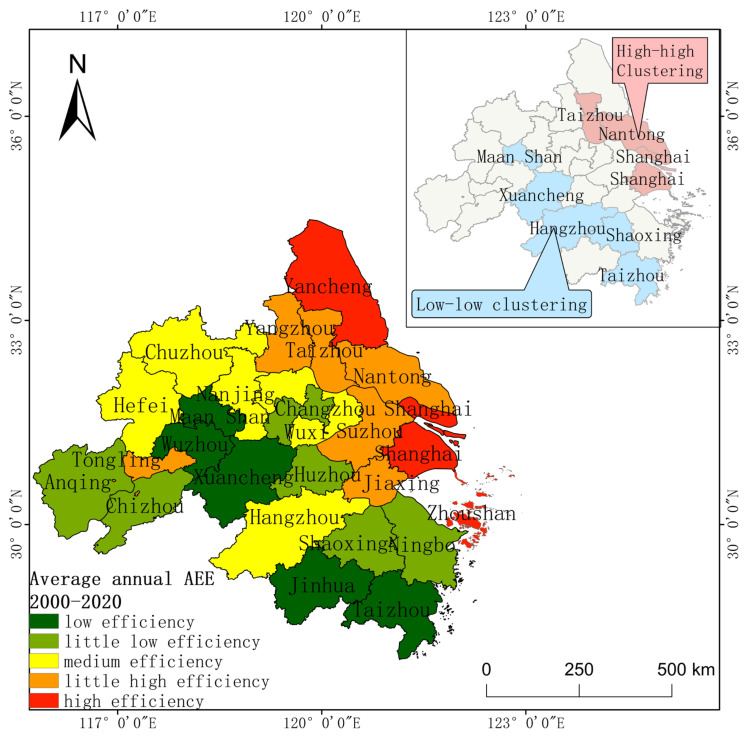
Annual average AEE and spatial distribution of clustered areas from 2000 to 2020.

**Table 1 ijerph-20-04786-t001:** AEE index system.

Indicator Type	Variables	Variable Description	Indicator Source
Inputs	Land Inputs	Total sown area of crops	Literature [5,43].
	Livestock input	Average annual rearing of pigs, cattle, sheep and poultry	Literature [5].
	Electricity input	Rural electricity consumption	Literature [20,43].
	Labor input	Employees in the primary industry	
	Machinery and power input	Total power of agricultural machinery	
	Agricultural material inputs	Agricultural fertilizer, agricultural diesel, pesticides and agricultural film	
Desired output	Agricultural output	Gross agricultural product	Literature [11,43].
	Agroecosystem services	Total value of ecosystem services	
Non-desired outputs	Carbon Emissions	Total agricultural carbon emissions	

**Table 2 ijerph-20-04786-t002:** Carbon emission factors of carbon emission sources.

Carbon Source Category	Carbon Source Subcategory	Carbon Emission Factor	Unit	Parameter Source
Land management	Fertilizer	0.8956	kgC/kg	Oak Ridge National Laboratory, USA
	Pesticides	4.9341	kgC/kg	
	Agricultural film	5.18	kgC/kg	Institute of Agricultural Resources and Ecological Environment, Nanjing
	Agricultural diesel	0.5927	kgC/kg	IPCC
	Tilling of farmland	312.6	kgC/km^2^	Institute of Biology and Technology, China Agricultural University
	Agricultural irrigation	266.48	kgC/hm^2^	Literature [11,41,51]
	Agricultural machinery power	0.18	kg/kw	
Livestock farming	Pigs	73.756	kgC/kg	IPCC Literature [45,47,50].
	Cattle	497.532	kgC/kg	
	Sheep	62	kgC/kg	
	Poultry	3.251	kgC/kg	
Crop cultivation	Soybeans	62.58021	kgC/hm^2^	Literature [48,51,52].
	Rice	19.50552	kgC/hm^2^	
	Corn	205.7832	kgC/hm^2^	
	Vegetables	342.1593	kgC/hm^2^	

**Table 3 ijerph-20-04786-t003:** Explanatory variable, index description, and symbol prognosis of AEE.

Variables	Variable Abbreviation	Index Description	Direction
Effective irrigation rate	Eir	Arable land area at year-end/effective irrigated area	Positive
Urbanization level	Url	Urban household population/regional year-end household population	Positive
Electricity consumption per unit of output value	Ecv	Rural electricity consumption/gross agricultural output	Unknown
Agricultural industry structure	Ais	Gross agricultural product/Gross agricultural output	Unknown
Crop cultivation structure	Ccs	Area of food crops sown/total area of crops sown	Unknown
Scale of cultivated land	Scl	Area under cultivation at the year-end/employed in agriculture	Positive
Agricultural mechanization intensity	Ami	Total power of agricultural machinery/total area of crops sown	Positive
Per capita income ratio between urban and rural areas	Pca	Per capita disposable income of urban residents/per capita disposable income of rural residents	Negative
Agricultural fertilizer application intensity	Afi	Disposable income	Negative

**Table 4 ijerph-20-04786-t004:** Global spatial correlation coefficients at different periods.

Year	2000–2005	2006–2010	2011–2015	2016–2020	2000–2020
Moran’s I	0.270969	0.361069	0.355739	0.038233	0.508774
Z value	2.481857	3.222867	3.152146	0.639974	4.389488
*p* value	0.013070	0.001269	0.001621	0.522190	0.000011

**Table 5 ijerph-20-04786-t005:** Regression results of factors influencing AEE from 2000 to 2020.

Variables	Coefficient Coe		*p* > |z|	
	Tobit	GWR	Tobit	GWR
Eir	0.0293402	0.2752174	0.498	0.001
Url	0.4851923	0.1983217	0.000	0.000
Ecv	−0.038422	−0.0332305	0.000	0.000
Ais	0.2207186	0.3928892	0.002	0.000
Ccs	0.2063777	0.1216250	0.001	0.000
Scl	0.0863592	0.1226107	0.000	0.000
Ami	0.042512	0.0467646	0.004	0.000
Pca	−0.0476017	−0.0300890	0.001	0.003
Afi	−0.1660607	−0.0283201	0.002	0.000

## Data Availability

The data that support the findings of this study are available from the corresponding author (L.Z.) upon justifiable request.

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
