# Peer review of "Spatio-Temporal Difference in Agricultural Eco-Efficiency and Its Influencing Factors Based on the SBM-Tobit Models in the Yangtze River Delta, China"

_ijerph, 2023, doi:10.3390/ijerph20064786_

Round 1

Reviewer 1 Report

Dear authors, this manuscript is interesting. 

Please, follows these recommendations: 

Please, improve size for Table 2 line 156.

Please, improve size and structure for Table 3 line 185. 

Please, add associated data to Figure 2 line 258 to better understand the graph. 

Please, improve Table 4 line 290. 

Please, insert a space between lines 307-308.

Please, read and consult recent publications on the topic analyzed. 

The authors analyzed the eco-efficiency in agriculture (EEE) on the basis of the carbon emissions system in spatial and temporal characteristics. They matched environmental data and spatial data.

It is a very important topic for the agricultural sector and because it highlights the associated environmental impacts. However, some studies have already been published in the past on the topics GIS and agriculture. Authors should enrich the list of references.

Compared to the other publications, the use of the TOBIT model for data analysis is interesting. It therefore presents a multidisciplinary approach.

Authors should graphically enhance the tables and associate data with Figure 2. Methodologically, they should state the hypotheses tested and verify the results.

The conclusions are clear but explain better the expression contained in line 503.

The authors must add other references from European and Western studies.

Authors need to improve table sizes and structures.

Reviewer 2 Report

This study conducts a regional study of AEE to reduce agricultural environmental pollution, improve the rationalization of agricultural production layout, and promote the realization of low-carbon goals in the Yangtze River Delta region. The work is in the scope of the journal, however, redaction and structure should be improved as indicated below, especially the methods should be clearer; the author is recommended to identify and practice sophisticated objectives for a journal publication. The author must justify the following points:

Comment 1: Several of the used references are outdated, which could negatively influence the novelty of the study. Hence, a deep analysis of recent scientific papers covering only the topic and leading to the submission hypothesis based on the gap analysis of the previously published research is required. The paper should be revised to highlight novelties. Please state the novelty of this paper in answer to the comments. Do we have existing studies on an exactly similar topic? What’s the specific contribution relevant to existing similar studies? Please provide some references related to similar existing studies and then the authors could state the contribution clearly in the manuscript.

Comment 2: The author is using a lot of abbreviations. Hence it is suggested to include a nomenclature at the beginning of the article.

Comment 3: The proposed approach in the section Materials and Methods is not outlined with the necessary vigor. The author needs to include sufficient methodological details in the paper and elaborate on the produced results from the proposed methods. Some sections must be added and others need to be relocated and rewritten to make it clearer for the readers. It is not clear to me if this work was built based on a single case study only, or if it was built based on a proposed approach validated within a case study.

Comment 4: The models and equations utilized are kind of simple. There exist more advanced models that could be utilized for better assessments. Why the author utilized these simplified models? Hence, the Mathematical Model presented in this work should be rewritten and better structured, including a clear explanation for each equation. Symbols for variables, marks, labels, etc. must be identical in the text, equations, figures, tables, and nomenclature. Variables must be in italics style.

Comment 5: In the Discussion Section, the author must differentiate between the results of this work based on; the proposed approach; and the applied case study building. This section should be improved by including a clear and concise analysis of all results presented in the previous section.

Comment 6: The Conclusion section is missing some necessary details. For example, the author needs to highlight the novelty and the materials and methods used in this work. Then the author should present the results of this work. Eventually, a summary of the limitations of this research as well as a recommendation for future works should be indicated.  

Reviewer 3 Report

The paper is interesting, but it needs to be improved further.

Introduction

L53-Other references at international level, may be added.

L138-"DEA model method"-More information of the DEA model used is needed and regarding its implementation. Also you must mention what is the orentation of the model? (input? output?). Present the equations of the DEA model.

4. Discussion-Well presented and quite complete.

Reviewer 4 Report

1The literature is not profound. The importance of this topic is not better put forward. Therefore, I hold that a separate literature should be written.

2、The topic of AEE is relatively old, I can not understand what the literature contribution is ? The change of study area is not innovative.

3、The selection of index system regarding the AEE should give ample literature reference. Moreover, the meanings of each indicator needs to be explained.

4、If there is spatial correlation of AEE across cities in YRD, the spatial Durbin model or GWR is more suitable. 

5、The language of this manuscript is relatively poor, I can not understand some sentence. Thus, I suggest that the manuscript needs to be polished under the native English speakers.

Round 2

Reviewer 4 Report

Thank authors for their revision. I agree with the publication.